# GS³LAM: Gaussian Semantic Splatting SLAM

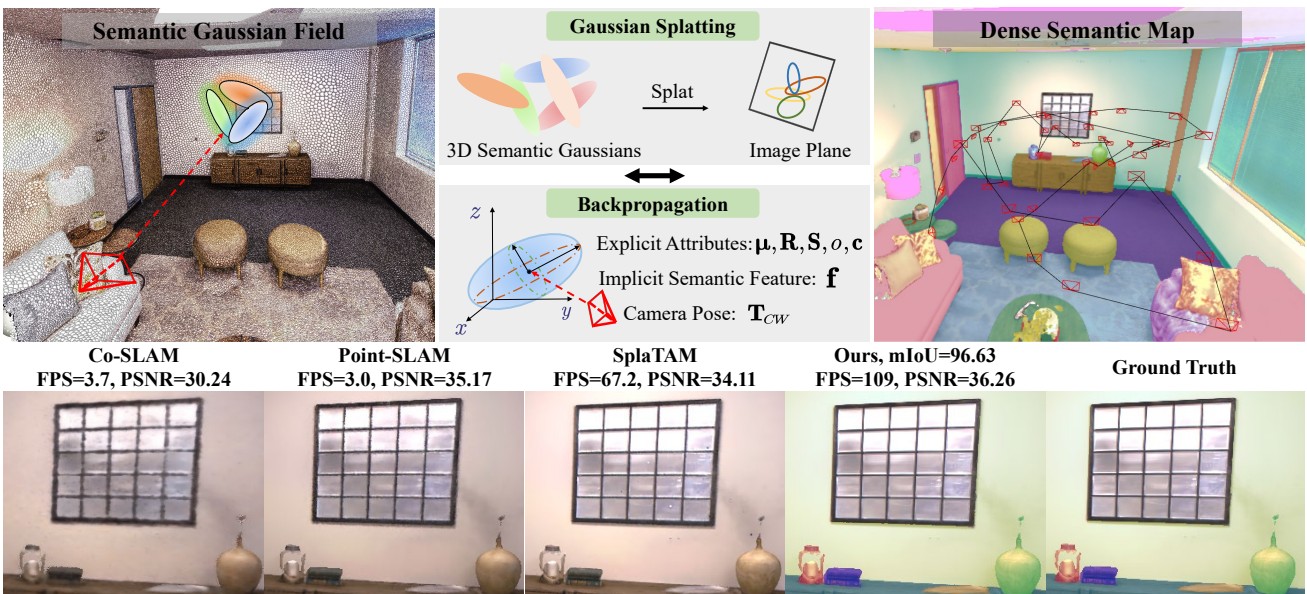

**Figure 1: Our proposed GS³LAM utilizes the 3D semantic Gaussian representation and the differentiable splatting rasterization pipeline, and jointly optimizes camera poses and field for appearance, geometry and semantics, achieving robust tracking, real-time high-quality rendering, and precise 3D semantic reconstruction.**

## ABSTRACT

Recently, the multi-modal fusion of RGB, depth, and semantics has shown great potential in the domain of dense Simultaneous Localization and Mapping (SLAM), as konwn as dense semantic SLAM. Yet a prerequisite for generating consistent and continuous semantic maps is the availability of dense, efficient, and scalable scene representations. To date, existing semantic SLAM systems based on explicit scene representations (points/meshes/surfels) are limited by their resolutions and inabilities to predict unknown areas, thus failing to generate dense maps. Contrarily, a few implicit scene representations (Neural Radiance Fields) to deal with these problems rely on time-consuming ray tracing-based volume rendering technique, which cannot meet the real-time rendering requirements of SLAM. Fortunately, the Gaussian Splatting scene representation has recently emerged, which inherits the efficiency and scalability of point/surfel representations while smoothly represents geometric structures in a continuous manner, showing promise in addressing the aforementioned challenges. To this end, we propose **GS³LAM**, a Gaussian Semantic Splatting SLAM framework, which takes multimodal data as input and can render consistent, continuous dense semantic maps in real-time. To fuse multimodal data, GS³LAM models the scene as a Semantic Gaussian Field (SG-Field), and jointly optimizes camera poses and the field by establishing error constraints between observed and predicted data. Furthermore, a Depth-adaptive Scale Regularization (DSR) scheme is proposed to tackle the problem of misalignment between scale-invariant Gaussians and geometric surfaces within the SG-Field. To mitigate the forgetting phenomenon, we propose an effective Random Sampling-based Keyframe Mapping (RSKM) strategy, which exhibits notable superiority over local covisibility optimization strategies commonly utilized in 3DGS-based SLAM systems. Extensive experiments conducted on the benchmark datasets reveal that compared with state-of-the-art competitors, GS³LAM demonstrates increased tracking robustness, superior real-time rendering quality, and enhanced semantic reconstruction precision. To make the results reproducible, the source code will be publicly released.

## CCS CONCEPTS

• **Computing methodologies → Reconstruction**.

## KEYWORDS

Semantic SLAM, Gaussian splatting, 3D segmentation

# 1 INTRODUCTION

By integrating semantic understanding into map, semantic Simultaneous Localization and Mapping (SLAM) achieves simultaneous estimation of camera poses while constructing maps that maintain consistency across geometry, appearance, and semantics. In comparison to conventional SLAM techniques, it excels in the identification, classification, and correlation of entities within scenes. Nowdays, semantic SLAM systems have been applied in various domains, such as robotics [1, 13] and autonomous driving [1, 10, 20].

To date, existing semantic SLAM systems based on explicit scene representations often resort to points/surfels [15, 23, 26, 28], grids [16], or voxels [8, 11, 17] to construct maps. Although these representations offer advantages in geometry, storage, computational efficiency, and scalability, they face challenges in predicting unknown regions and are constrained by limited resolutions, thus being unable to generate dense semantic maps. Contrarily, recent emerging neural rendering techniques based on implicit scene representations, such as Neural Radiance Fields (NeRF) [14], have shown potentials to deal with these challenges. NeRF portrays scenes as continuous implicit volume functions, enabling realistic novel view synthesis with minimal storage requirements. Based on it, several studies [3, 32] incorporate additional MLP channels to encode and decode semantic labels, while jointly optimizing camera poses and semantic scenes. However, due to the computationally expensive ray tracing-based volume rendering technique of NeRF, these methods fail to meet the real-time demands of SLAM.

Fortunately, we observe the emergence of 3D Gaussian Splatting (3DGS) [7], which demonstrates exceptional capabilities in dense 3D reconstruction. This method represents the scene as dense Gaussian clouds and achieves efficient rendering through tile-based rasterization. We show that 3DGS has great potential in addressing the aforementioned challenges. As a semantic SLAM scene representation, it inherits the efficiency, locality, and modifiability of point/surfel representations while smoothly and differentially representing the geometric structure in a continuous manner, enabling the reconstruction of rich and complex details in dense maps. To further improve the capabilities of semantic SLAM in tracking, rendering, and semantic reconstruction, it is a natural idea to extend 3DGS as a semantic scene representation, but surprisingly such a simple idea has seldom been explored in existing literature. In this work, based on the above-mentioned findings, we propose a dense semantic SLAM framework, **GS³LAM** (**G**aussian **S**emantic **S**platting **SLAM**), to fully leverage the advantages of 3DGS.

However, the effective embedding and real-time optimization of high-dimensional semantic categories pose profound challenges for GS³LAM. To deal with these issues, GS³LAM models the scene as a Semantic Gaussian Field (SG-Field), wherein semantic categories are represented as low-dimensional implicit features. By means of a simple decoder, GS³LAM efficiently transforms these features into semantic categories, facilitating the conversion between 3D implicit features and 2D semantic labels.

Furthermore, within the SG-Field, irregular Gaussian scales hinder the accurate representation of geometric surfaces, making it unacceptable for pixel-level semantic reconstruction. To address this issue, we propose a Depth-adaptive Scale Regularization (DSR) strategy. This strategy constrains scales within a depth-dependent

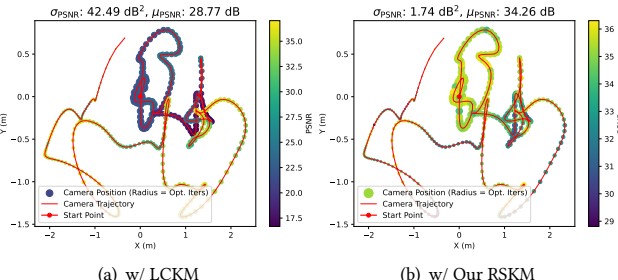

(a) w/ LCKM      (b) w/ Our RSKM

**Figure 2: Illustration of optimization bias on Replica "Office 3".**

range, indirectly aligning Gaussians with geometric surfaces, thus can effectively reduce blurring on object surfaces and enhance both tracking robustness and semantic reconstruction accuracy.

Finally, to address the forgetting phenomenon in GS³LAM, we propose a Random Sampling-based Keyframe Mapping (RSKM) strategy, which proves to be more effective than the Local Covisibility Keyframe Mapping (LCKM) strategy commonly adopted in 3DGS-based SLAM systems. Our observation suggests that the latter method introduces a considerable bias during the optimization of the Gaussian field, thereby leading to poor global map consistency. In particular, as depicted in Fig. 2(a), frames with dense co-observations (dense camera trajectories) and increased optimization iterations (large point radii) exhibit lower PSNR values (darker color), suggesting challenges in achieving convergence of the Gaussian field under the LCKM strategy. Conversely, as shown in Fig. 2(b), our proposed RSKM strategy not only enhances the rendering quality of the global map (higher mean PSNR, $\mu_{PSNR}$ ) but also ensures high consistency among all perspectives (smaller PSNR variance, $\sigma_{PSNR}$), effectively reducing the optimization bias.

Our contributions are summarized as follows:

(1) As depicted in Fig. 1, **GS³LAM** is a **G**aussian **S**platting **S**emantic **SLAM** framework, which models the scene as a Semantic Gaussian Field (SG-Field) to efficiently facilitate the conversion between 3D semantic features and 2D labels. By the joint optimization of camera poses and field for appearance, geometry, and semantics, it achieves robust tracking, real-time high-quality rendering, and precise semantic reconstruction.

(2) A Depth-adaptive Scale Regularization (DSR) scheme is proposed to reduce the blurring of geometric surfaces induced by irregular Gaussian scales within the SG-Field. By constraining Gaussian scales within a reasonable range determined by depth, it alleviates the ambiguity of geometric surfaces, thereby enhancing accuracy in semantic reconstruction.

(3) To address the forgetting phenomenon in GS³LAM, we propose an efficacious Random Sampling-based Keyframe Mapping (RSKM) strategy, which exhibits notable superiority over prevalent local covisibility optimization strategies commonly employed in 3DGS-based SLAM systems. As shown in Fig. 2, our method significantly enhances both the reconstruction accuracy and rendering quality while maintaining the global consistency of the semantic map.

(4) Extensive experiments conducted on Replica [21] and ScanNet [2] datasets demonstrate that our GS³LAM outperforms its counterparts in terms of tracking accuracy, rendering quality and speed, and semantic reconstruction.

## 2 RELATED WORK

### 2.1 Scene Representation for Semantic SLAM

Semantic SLAM systems typically utilize various scene representations such as points/surfels [15, 23, 26, 28], grids [16], or voxels [8, 11, 17] to facilitate the creation of semantic maps. For instance, the mesh-based SLAM++ [18] models the world as a graph, with each node capturing an estimated $SE(3)$ pose, and represents each 3D object as a mesh. Another notable system, Kimera [1], annotates semantic labels onto the faces of meshes, enabling the real-time construction of metric-semantic 3D mesh environment models. The surfel-based SemanticFusion [13], on the other hand, builds upon real-time ElasticFusion [28] and utilizes CNN predictions of pixel categories and Bayesian update schemes to track the category probability distribution of each surfel, thereby establishing a globally consistent semantic map. Despite the benefits that these representation techniques present in terms of geometry, storage, computational efficiency, and scalability, they encounter difficulties in predicting unexplored areas and are restricted by limited resolutions, thereby incapable of producing dense semantic maps.

### 2.2 NeRF-based and 3DGS-based SLAM

In recent years, neural rendering techniques based on continuous scene representations, such as Neural Radiance Fields (NeRF) [14] and 3D Gaussian Splatting (3DGS) [7], have emerged, showing significant potential in photorealistic rendering and dense reconstruction. NeRF represents scenes as continuous implicit volume functions, enabling realistic novel view synthesis with modest storage requirements. For NeRF-based SLAM, existing methods can be categorized into two main types, implicit (MLP-based) representation methods and hybrid representation methods. The MLP-based iMAP [24] is the first to employ neural radiance for tracking and mapping tasks, offering memory-efficient dense map representations but failing to scale to large scenes. On the other hand, hybrid representation methods combine the scalability of explicit representations with the low memory consumption of implicit representations, significantly improving scene scalability and accuracy. For instance, NICE-SLAM [34] proposes hierarchical multi-feature grids, Co-SLAM [25] adopts multi-resolution hash grids, and Vox-Fusion [30] utilizes octrees for dynamic map expansion. ESLAM [5] and Point-SLAM [19], in addition, employ tri-planes and neural point clouds respectively for volume rendering, significantly enhancing mapping capabilities. Furthermore, some methods [9, 32] incorporate additional MLP channels to encode and decode semantic labels, while optimizing camera poses and semantic scenes simultaneously. However, due to the computational expense of NeRF's ray-tracing-based volume rendering, these methods fail to meet the real-time requirements of SLAM.

In contrast to NeRF, 3DGS achieves remarkable capabilities by representing scenes as dense Gaussian clouds and a tile-based rasterization, thereby accomplishing high-quality and efficient rendering. Recently, several SLAM methods [4, 6, 12, 29] based on 3DGS have been developed. They represent scenes as 3D Gaussians and directly backpropagate to optimize camera poses and the Gaussian fields.

Notably, the aforementioned NeRF-based and 3DGS-based SLAM predominantly focus on the optimization of RGB maps, which poses limitations on their direct applicability to downstream tasks such as navigation and localization. To address this challenge, we propose an approach that integrates semantic features into 3D Gaussians and jointly optimizes camera poses, geometry, appearance, and semantics, enabling accurate 3D semantic mapping, robust camera tracking, and high-quality real-time rendering.

## 3 METHODOLOGY

### 3.1 Framework Overview

As illustrated in Fig. 3, our GS³LAM framework is designed to process RGB-D data with unknown camera poses and corresponding 2D semantic labels. It models the scene as a SG-Field, wherein each 3D Gaussian is characterized by its position $\mu$, rotation matrix $\mathbf{R}$, scaling matrix $\mathbf{S}$, opacity $o$, color $\mathbf{c}$, and semantic feature $\mathbf{f}$. To facilitate progressive reconstruction of semantic maps with geometric-semantic consistency, we employ an adaptive 3D Gaussian expansion technique and propose the RSKM strategy to alleviate the forgetting phenomenon. Finally, GS³LAM optimizes camera poses and the SG-Field using appearance, geometry, and semantics, along with the proposed DSR scheme which ensures the alignment between geometry and semantics within the field.

### 3.2 Semantic Gaussian Field

Our goal is to establish a scene representation that efficiently captures the geometry, appearance, and semantics of the scene, thereby facilitating the production of realistic dense map and precise semantic reconstruction. To accomplish this objective, we model the scene as a SG-Field $\mathcal{G}$ containing $N$ semantic Gaussians,

$$\mathcal{G} := \{(\mu_i, \Sigma_i, o_i, \mathbf{c}_i, \mathbf{f}_i) | i = 1, 2, \ldots, N\}, \quad (1)$$

where the $i$-th 3D semantic Gaussian is defined by its position $\mu_i \in \mathbb{R}^3$, covariance matrix $\Sigma_i \in \mathbb{R}^{3 \times 3}$, opacity $o_i \in \mathbb{R}$, RGB color $\mathbf{c}_i \in \mathbb{R}^3$, and semantic feature $\mathbf{f}_i \in \mathbb{R}^{N_{sem}}$ ($N_{sem}$ denotes the number of objects in the field). To optimize the parameters of the SG-Field using gradient descent, the covariance matrix $\Sigma_i$ can be represented equivalently as [7],

$$\Sigma_i = \mathbf{R}_i \mathbf{S}_i \mathbf{S}_i^T \mathbf{R}_i^T, \quad (2)$$

where $\mathbf{S}_i \in \mathbb{R}^{3 \times 3}$ represents a diagonal scaling matrix, and $\mathbf{R}_i \in \mathbb{R}^{3 \times 3}$ denotes a rotation matrix.

#### 3.2.1 *Color and Depth Splatting-Rendering.* When provided with an optimized SG-Field $\mathcal{G}$, along with a world-to-camera viewing transformation (also known as the camera pose) $\mathbf{T}_{CW} \in \mathbb{R}^{4 \times 4}$, the $i$-th 3D semantic Gaussian can be projected onto the 2D image plane for rendering with a $2 \times 2$ covariance matrix $\Sigma_i^{2D}$ [35],

$$\Sigma_i^{2D} = \mathbf{J}_i \mathbf{R}_{CW} \Sigma_i \mathbf{R}_{CW}^T \mathbf{J}_i^T, \quad (3)$$

where $\mathbf{J}_i \in \mathbb{R}^{2 \times 3}$ is the Jacobian of the $i$-th Gaussian centroid projected onto the 2D image plane with respect to its position in the camera coordinate system, and $\mathbf{R}_{CW} \in \mathbb{R}^{3 \times 3}$ denotes the rotation matrix of the camera pose $\mathbf{T}_{CW}$. Upon the projection of 3D Gaussians onto the image plane, the color of a single pixel $\hat{\mathbf{c}}_{pix}$ is rendered by sorting the Gaussians in depth order and performing front-to-back $\alpha$-blending rendering as,

$$\hat{\mathbf{c}}_{pix} = \sum_i^M \mathbf{c}_i \alpha_i \prod_j^{i-1} (1 - \alpha_j), \quad (4)$$

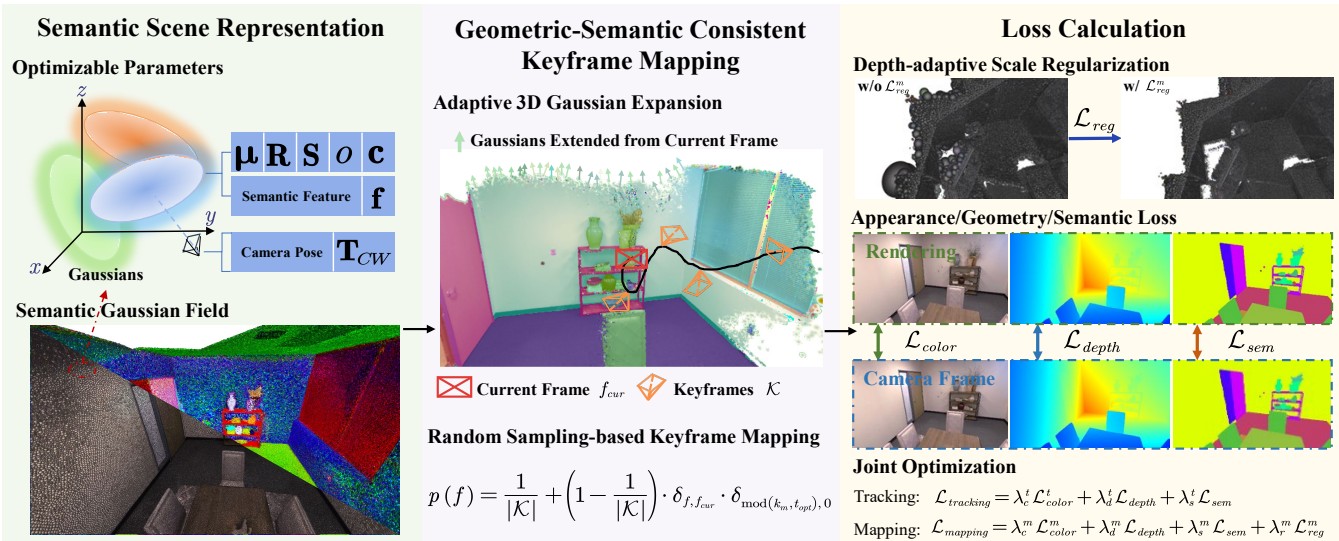

**Figure 3: The framework overview of GS³LAM. GS³LAM models the scene as a Semantic Gaussian Field (SG-Field). For geometric-semantic consistent keyframe mapping, an adaptive 3D Gaussian expansion technique and a Random Sampling-based Keyframe Mapping (RSKM) strategy are employed. GS³LAM optimizes camera poses and SG-Field using appearance, geometry, and semantics, along with a Depth-adaptive Scale Regularization (DSR) scheme.**

where $M$ is the number of sorted Gaussians overlapping with the given pixel. The density $\alpha_i$ is computed from the 2D covariance matrix $\Sigma_i^{2D}$ and the opacity $o_i$ of the $i$-th 3D Gaussian as,

$$\alpha_i = o_i \cdot \exp(-\frac{1}{2} \boldsymbol{\sigma}_i^T (\Sigma_i^{2D})^{-1} \boldsymbol{\sigma}_i), \tag{5}$$

where $\boldsymbol{\sigma}_i \in \mathbb{R}^2$ is the offset between the pixel center and the $i$-th projected 2D Gaussian center. Likewise, the depth $\hat{d}_{pix}$ of a single pixel is rendered by,

$$\hat{d}_{pix} = \sum_i^M d_i \alpha_i \prod_j^{i-1} (1 - \alpha_j), \tag{6}$$

where $d_i$ is the depth of the $i$-th Gaussian centroid with respect to the camera coordinate system.

### 3.2.2 *Semantic Feature Splatting-Rendering and Decoding.*
To develop a versatile pipeline for embedding semantic features, it is imperative that our approach possesses the capability to generate semantic feature maps of varying sizes and dimensions. To fulfill this requirement, we employ a rendering pipeline based on differentiable 3DGS framework similar to color and depth. Specifically, the 2D semantic feature of a single pixel $\hat{\mathbf{f}}_{pix}$ can be rendered as,

$$\hat{\mathbf{f}}_{pix} = \sum_i^M \mathbf{f}_i \alpha_i \prod_j^{i-1} (1 - \alpha_j), \tag{7}$$

where $\mathbf{f}_i$ denotes the $N_{sem}$-dimensional semantic feature vector of the $i$-th 3D Gaussian. To decode discrete semantic labels from continuous 2D semantic features, we initially utilize a CNN decoder $\mathcal{F}_{cnn}$ to restore the low-dimensional feature to $K_{sem}$ dimensions ($K_{sem}$ represents the semantic label categories). Then, a softmax classification is employed on the high-dimensional feature to obtain

the semantic label $\hat{s}_{pix}$ of a single pixel,

$$\hat{s}_{pix} = softmax(\mathcal{F}_{cnn}(\hat{\mathbf{f}}_{pix})). \tag{8}$$

Due to $N_{sem} \ll K_{sem}$, GS³LAM can efficiently achieve the conversion between 3D semantic features and 2D semantic labels, seamlessly embedding semantic features into 3DGS-based SLAM while maintaining the optimization efficiency.

### 3.2.3 *Decoupled Optimization.*
In our GS³LAM system, the parameters to be optimized include $P$ camera poses $\mathcal{T}$ and the SG-Field $\mathcal{G}$,

$$\Theta_{\mathcal{T}} := \{(\mathbf{q}_i, \mathbf{t}_i)\}_{i=1}^P, \quad \Theta_{\mathcal{G}} := \{\{(\boldsymbol{\mu}_i, \Sigma_i, o_i, \mathbf{c}_i, \mathbf{f}_i)\}_{i=1}^N, \mathcal{F}_{cnn}(\cdot)\}, \tag{9}$$

where $\mathbf{q}_i = [q_i^w, q_i^x, q_i^y, q_i^z]^T$ represents the rotation quaternion, $\mathbf{t}_i = [t_i^x, t_i^y, t_i^z]^T$ denotes the translation vector, and the parameters of the SG-Field $\mathcal{G}$ are defined in Eq. (1) and Eq. (8). Simultaneously optimizing both the camera pose parameters $\Theta_{\mathcal{T}}$ and the semantic Gaussian parameters of SG-Field $\Theta_{\mathcal{G}}$ is time-consuming and challenging. Therefore, a strategy of decoupling the optimization of camera poses and field parameters is adopted. In the tracking stage (Sec. 3.4), GS³LAM optimizes the camera pose of the current frame $\mathbf{T}_t$ with reference to a pre-trained SG-Field $\mathcal{G}_{t-1}$. During the mapping phase (Sec. 3.3), it optimizes the current SG-Field $\mathcal{G}_t$ based on accurately estimated camera poses $\mathbf{T}_0, \mathbf{T}_1, \ldots, \mathbf{T}_t$.

## 3.3 Geometric-Semantic Consistent Mapping

### 3.3.1 *Adaptive 3D Gaussian Expansion.*
To accommodate to the paradigm of incremental reconstruction in SLAM, an adaptive 3D Gaussian expansion strategy is employed during the mapping process. Following the tracking of a frame, we re-render the current frame and compute the cumulative opacity $\hat{o}_{pix}$ for each pixel. This process can be seamlessly integrated into the differentiable

rasterization pipeline of 3DGS [7],

$$\hat{o}_{pix} = \sum_i^M \alpha_i \prod_j^{i-1} (1 - \alpha_j). \tag{10}$$

Inspired by [6, 29], cumulative opacity and depth are employed to construct a mask for the unobservable regions of the SG-Field $\mathcal{G}_{t-1}$ under the viewpoint $\mathbf{T}_t$ in the current frame,

$$M_{unobs} = \mathbb{I}(\hat{o}_{pix} < \tau_{unobs}) \vee \mathbb{I}(\hat{d}_{pix} > d_{gt} \wedge L_1(d_{gt}, \hat{d}_{pix}) > 50\tilde{L}_1(d_{gt}, \hat{d}_{pix})), \tag{11}$$

where $\mathbb{I}$ denotes the indicator function, $\tau_{unobs}$ represents cumulative opacity threshold for unobservable regions, and $\tilde{L}_1(d_{gt}, \hat{d}_{pix})$ refers to the median of the $l_1$-norm error between the observed depth $d_{gt}$ and the rendered depth $\hat{d}_{pix}$. This mask indicates regions characterized by inadequate map density ($\hat{o}_{pix} < \tau_{unobs}$), or where additional geometry is anticipated to exist ahead of the presently estimated geometry ($L_1(d_{gt}, \hat{d}_{pix}) > 50\tilde{L}_1(d_{gt}, \hat{d}_{pix})$). Relying on this mask, we can dynamically and adaptively integrate newly observed regions into the SG-Field ($\mathcal{G}_{t-1} \xrightarrow{M_{unobs}} \mathcal{G}_t$). Concurrently, this mask serves to prevent the addition of new Gaussians to areas where the current Gaussian adequately represents the scene geometry, thereby effectively managing the number of Gaussians within $\mathcal{G}_t$, leading to decreased memory usage and optimization time.

### 3.3.2 *Depth-adaptive Scale Regularizationn (DSR)*.
Based on the mask $M_{unobs}$ of the current frame, all unobservable pixels are used to expand new semantic Gaussians. Specifically, for each pixel, we add a new semantic Gaussian with the color of that pixel, the semantic feature represented by random $N_{sem}$-dimensional Spherical Harmonics coefficients, the centroid at the location of the unprojection of that pixel depth $d_{gt}$, an opacity of 0.5, and scales initialized to $d_{gt}/f$, where $f$ denotes the camera focal length. Although this scaling initialization strategy shows higher efficiency compared to the KNN method in 3DGS [7], the variations in depth range across different frames result in significant variance in the 3D Gaussian scales corresponding to these frames. Such variance is not conducive to the SG-Field optimization. Furthermore, this strategy fails to adaptively represent high- and low-frequency information within the field, i.e., using smaller scales in high-frequency regions and larger scales in low-frequency regions. To address these challenges, we propose a depth-adaptive scale regularization term,

$$\mathcal{L}_{big} = \frac{\sum_i s_i \mathbb{I}(s_i > s_{big})}{\sum_i \mathbb{I}(s_i > s_{big})}, \quad \mathcal{L}_{small} = \frac{\sum_i -\log(s_i)\mathbb{I}(s_i < s_{small})}{\sum_i \mathbb{I}(s_i < s_{small})}, \tag{12}$$

where $s_i$ denotes the scale of the $i$-th Gaussian, $s_{big}$ and $s_{small}$ adhere the $2\sigma$ rule, i.e., $s_{big} = \mu_s + 2\sigma_s$ and $s_{small} = \mu_s - 2\sigma_s$. These terms constrain the global Gaussian scales within a reasonable range ($\mu_s - 2\sigma_s < s < \mu_s + 2\sigma_s$), thereby preventing excessively large or small Gaussians. Moreover, they indirectly align the Gaussians with geometric surfaces, reducing the blurriness of object edges, and achieving spatial alignment between geometry and semantics.

### 3.3.3 *Random Sampling-based Keyframe Mapping (RSKM)*.
As illustrated in Fig. 4, due to the optimization properties of 3DGS [7], 3DGS-based SLAM systems inherently demonstrate a propensity for forgetting during the incremental reconstruction procedure. In order to alleviate this issue, SplaTAM [6] and MonoGS [12] adopt a strategy of Local Co-visible Keyframe Mapping (LCKM),

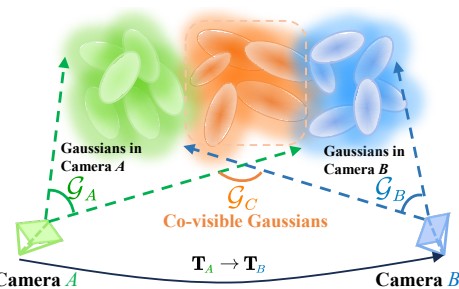

**Figure 4: The forgetting problem in SG-Field. During the incremental optimization process, Gaussians $\mathcal{G}_A$ in camera $A$ are initially optimized. However, when optimizing the Gaussians $\mathcal{G}_B$ in camera $B$, the co-visible Gaussians $\mathcal{G}_C = \mathcal{G}_A \cap \mathcal{G}_B$ tend to be excessively fitted to the latest frame of camera $B$, resulting in a decrease in the reconstruction quality of the previous frame captured by camera $A$.**

wherein, during the optimization of the current frame, the remaining keyframes co-visible with the current frame are selected to participate in optimization together. However, as shown in Fig. 2, we observe that this approach leads to under-optimized regions with sparse co-visibility, while areas with numerous co-visibility frames exhibit a tendency towards convergence difficulties, resulting in significantly biased semantic maps. To address this problem, we propose the RSKM strategy, which effectively reduces the optimization bias and enhances the global consistency of the SG-Field.

In the process of mapping the current frame $f_{cur}$, during each iteration, RSKM selects a frame $f$ in the keyframe set $\mathcal{K}$ with probability $p(f)$ to participate in the optimization,

$$p(f) = \frac{1}{|\mathcal{K}|} + \left(1 - \frac{1}{|\mathcal{K}|}\right) \cdot \delta_{f, f_{cur}} \cdot \delta_{\text{mod}(k_m, t_{opt}), 0}, \tag{13}$$

where $|\mathcal{K}|$ denotes the size of the keyframe set $\mathcal{K}$, $k_m$ represents the number of iterations for mapping, $\delta_{i,j}$ is the Kronecker delta function, which equals 1 if $i = j$ and 0 otherwise, and the optimization target interval $t_{opt}$ is used to balance optimization between the current frame and keyframes. It is noteworthy that RSKM does not involve time-consuming keyframe selection operations as done in SplaTAM [6] and Point-SLAM [19], yet still achieves a high level of effectiveness in ensuring the global consistency of semantic maps.

### 3.3.4 *Objective Functions*.
Based on the aforementioned sampling strategy, the optimization objective of SG-Field $\mathcal{G}_t$ can be defined as,

$$\mathcal{G}_t^* = \arg\min_{\Theta_\mathcal{G}} \sum_i^{k_m} \mathcal{L}_{mapping} (\mathcal{R}(\mathbf{T}_i \odot \mathcal{G}_t), O_i), \tag{14}$$

where $k_m$ represents the number of iterations for mapping, $\mathcal{L}_{mapping}$ refers to the mapping loss, $\mathbf{T}_i$ and $O_i$ represent the camera pose and the ground truth data (RGB image, depth map and semantic labels) of the associated frame respectively, $\mathcal{R}$ denotes rasterization rendering, and $\odot$ represents the transformation of $\mathcal{G}_t$ with $\mathbf{T}_i$.

To ensure the consistency of multimodality within $\mathcal{G}_t$, $\mathcal{L}_{mapping}$ encompasses constraints related to appearance, semantics, geometry, and geometric-semantic spatial alignment. The color loss $\mathcal{L}_{color}^m$ is an $l_1$ loss combined with a D-SSIM [27] term,

$$\mathcal{L}_{color}^m = (1-\lambda) \|\hat{\mathbf{c}}_{pix} - \mathbf{c}_{gt}\|_1 + \lambda (1 - \text{D-SSIM}(\hat{\mathbf{c}}_{pix}, \mathbf{c}_{gt})), \tag{15}$$

where $\hat{\mathbf{c}}_{pix}$ and $\mathbf{c}_{gt}$ denote the rendered and observed color, and we use $\lambda = 0.2$ in all our tests. A binary cross entropy (BCE) loss is applied as the semantic loss,

$$\mathcal{L}_{sem} = -\left(s_{gt} \cdot \log(\hat{s}_{pix}) + (1 - s_{gt}) \cdot \log(1 - \hat{s}_{pix})\right), \quad (16)$$

where $\hat{s}_{pix}$ is the decoded semantic label from the semantic feature, and $s_{gt}$ is the input semantic label provided by the dataset or generated using state-of-the-art semantic segmentation models. An $l_1$ depth loss is utilized to guide geometry,

$$\mathcal{L}_{depth} = \left\|\hat{d}_{pix} - d_{gt}\right\|_1, \quad (17)$$

where $\hat{d}_{pix}$ and $d_{gt}$ are rendered depth and ground truth depth. Finally, the inclusion of the regularization terms for the Gaussian scales from Eq. (12) constitutes the complete mapping loss $\mathcal{L}_{mapping}$,

$$\mathcal{L}_{mapping} = \lambda_c^m \mathcal{L}_{color}^m + \lambda_d^m \mathcal{L}_{depth} + \lambda_s^m \mathcal{L}_{sem} \\ + \lambda_{big}^m \mathcal{L}_{\text{big}} + \lambda_{small}^m \mathcal{L}_{\text{small}}, \quad (18)$$

where $\lambda_c^m$, $\lambda_d^m$, $\lambda_s^m$, $\lambda_{big}^m$, and $\lambda_{small}^m$ control the weight of each term.

### 3.4 Frame-to-Model Tracking

Given an optimized SG-Field $\mathcal{G}_{t-1}$, GS$^3$LAM employs the frame-to-model strategy to optimize the world-to-camera poses $\mathcal{T}$. In particular, for the first frame, the camera pose $\mathbf{T}_0$ is initialized as the identity matrix. Then, adhering to the methodology outlined in Sec. 3.3, all pixels are initialized as Gaussians, and the mapping process is executed for $k_{init}$ iterations to yield the initially optimized $\mathcal{G}_0$. When a new frame arrives, GS$^3$LAM initializes the camera pose $\mathbf{T}_t$ using the constant velocity assumption as,

$$\mathbf{T}_t = \mathbf{T}_{t-1} \mathbf{T}_{t-2}^{-1} \mathbf{T}_{t-1} \quad (19)$$

Then, the SG-Field $\mathcal{G}_{t-1}$ is transformed into the camera coordinate system via $\mathbf{T}_t$, which is optimized by minimizing the tracking loss $\mathcal{L}_{tracking}$ between the rendered $\mathcal{R}(\cdot)$ and the ground truth data $O$,

$$\mathbf{T}_t^* = \arg\min_{\Theta_\mathcal{T}} \mathcal{L}_{tracking}\left(\mathcal{R}\left(\mathbf{T}_t \odot \mathcal{G}_{t-1}\right), O\right). \quad (20)$$

It is noteworthy that during the aforementioned optimization process, all attributes of the SG-Field $\mathcal{G}_{t-1}$ are frozen, separating the camera movement from the deformation, densification, pruning, and self-rotation of the 3D Gaussian points.

It is apparent that the SG-Field $\mathcal{G}_{t-1}$ inadequately observes all regions within the current frame. To improve the robustness and stability of tracking, $\mathcal{L}_{tracking}$ is designed to be aware of observable and geometrically normal regions, jointly minimizing photometric, geometric, and semantic errors,

$$\mathcal{L}_{tracking} = M_{obs}\left(\lambda_c^t \mathcal{L}_{color}^t + \lambda_d^t \mathcal{L}_{depth} + \lambda_s^t \mathcal{L}_{sem}\right), \quad (21)$$

$$M_{obs} = \mathbb{I}(\hat{o}_{pix} > \tau_{obs}) \wedge \mathbb{I}\left(L_1(d_{gt}, \hat{d}_{pix}) < 10\tilde{L}_1(d_{gt}, \hat{d}_{pix})\right),$$

where $M_{obs}$ denotes the mask of well-optimized depth in observable regions ($\hat{o}_{pix} > \tau_{obs}$) of the SG-Field $\mathcal{G}_{t-1}$ under the viewpoint $\mathbf{T}_t$, which holds significant importance for tracking. $\mathcal{L}_{color}^t = \left\|\hat{\mathbf{c}_{pix}} - \mathbf{c}_{gt}\right\|_1$ solely employs the $l_1$ loss, and $\lambda_c^t$, $\lambda_d^t$, $\lambda_s^t$ modulate the weight of each term.

## 4 EXPERIMENT

### 4.1 Setup

#### 4.1.1 Implementation Details.
GS$^3$LAM was implemented in Python using the PyTorch framework, and trained on a workstation with an AMD EPYC 7302 16-Core Processor and an NVIDIA GeForce RTX 3090 GPU. For SG-Field, we employed isotropic Gaussians, with semantic feature dimension $N_{sem} = 16$ and semantic label categories $K_{sem} = 256$. Regarding the cumulative opacity mask, we set $\tau_{unobs} = 0.5$ for adaptive 3D Gaussian expansion and $\tau_{obs} = 0.99$ for observable region-aware tracking. As for RSKM, the optimization target interval $t_{opt}$ was set to 10 for mapping. For the initial frame of mapping, the number of iterations $k_{init}$ was set to 1000 for Replica [21] and 500 for Scannet [2].

#### 4.1.2 Datasets and Evaluation Metrics.
Following [6, 19, 30, 33, 34], we used 8 scenes from the virtual Replica [21] and 5 subsets of real-world ScanNet [2] for tracking and rendering quality comparison. Rendering quality was assessed utilizing objective metrics including Peak Signal-to-Noise Ratio (PSNR), SSIM [27], and LPIPS [31]. Tracking accuracy was quantified by the ATE RMSE [22]. Semantic segmentation performance was gauged using the mean Intersection over Union (mIoU). In all of our tables, best results are highlighted as first and second .

#### 4.1.3 Baseline Methods.
We conducted a comparative analysis between our proposed GS$^3$LAM and several state-of-the-art dense neural RGBD SLAM methodologies, including NICE-SLAM [34], Vox-Fusion [30], ESLAM [5], Co-SLAM [25] and Point-SLAM [19]. Additionally, we expanded our comparison to encompass leading 3DGS-based SLAM techniques, specifically SplaTAM [6] and GS-SLAM [29]. In the context of semantic reconstruction, our method underwent evaluation against NeRF-based NIDS SLAM [3], DNS SLAM [9] and SNI-SLAM [32].

### 4.2 Rendering Evaluation

Tables 1 and 2 present the comparative rendering results of GS$^3$LAM with state-of-the-art NeRF-based and 3DGS-based SLAM systems on the ScanNet [2] and Replica [21] datasets, respectively. The results demonstrate that GS$^3$LAM achieves the best performance across commonly used metrics. On the Replica dataset, our approach outperforms the runner-up methods Point-SLAM [19] and GS-SLAM [29] by 1.09 dB, 0.014, and 0.039 in terms of PSNR, SSIM and LPIPS, respectively. Moreover, on the real-world ScanNet dataset, our superiority is more pronounced, with our method surpassing Point-SLAM [19] by 3.04 dB in PSNR, 0.117 in SSIM, and 0.292 in LPIPS. Compared to the 3DGS-based SplaTAM [6] and GS-SLAM [29], the semantic embedding and DSR scheme in GS$^3$LAM enable the Gaussian model to focus more on the details of object edges and eliminate surface blurring. Additionally, our proposed RSKM strategy effectively addresses the challenge of convergence in regions with abundant covisibility, as well as the issue of sub-optimal optimization in regions with sparse covisibility, achieving a balance between local and global optimization. This approach effectively alleviates the forgetting phenomenon inherent in 3DGS-based SLAM, thereby facilitating globally consistent and realistic rendering performance. Qualitatively, the visualization results in Fig. 5 demonstrate that NeRF-based Co-SLAM [25] and Point-SLAM

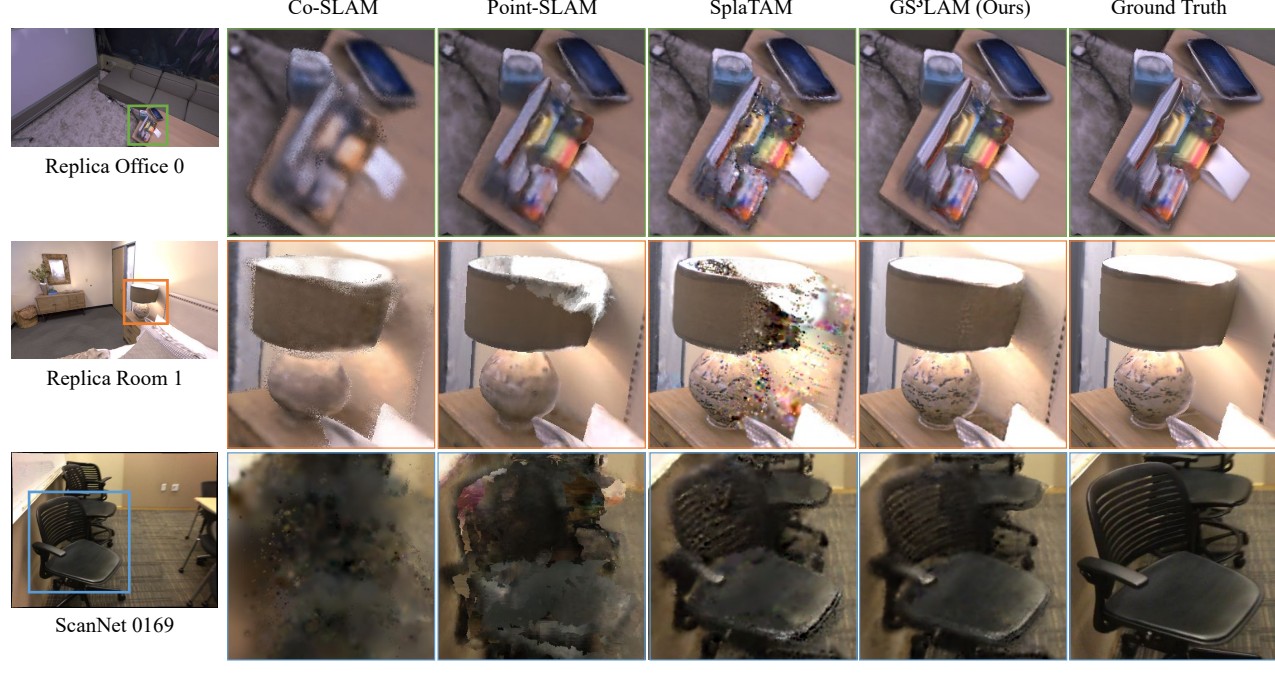

**Figure 5: Qualitative comparison with SOTA methods on virtual Replica [21] and real-world ScanNet [2] datasets.**

[19] exhibit inaccurate scene representations and are susceptible to lighting effects, leading to significant artifacts. SplaTAM [6] tends to get trapped in local optima, making convergence difficult or suboptimal, resulting in noticeable holes and blurring. In contrast, GS³LAM produces higher-quality and more realistic images with more structure details in both global and edge regions compared to other methods. It is noteworthy that, owing to the efficient semantic scene representation of SG-Field and the tile-based rasterization technology, GS³LAM achieves real-time rendering of RGB, depth, and semantics at 109.12 FPS on the 1200 × 680 Replica dataset, a 36.86-fold improvement over NeRF-based SLAM methods. Similarly, on the 640 × 480 ScanNet dataset, it reaches 499.78 FPS, providing possibilities for downstream real-time tasks. **More results can be found in the supplemental materials.**

**Table 1: Rendering performance on ScanNet [2].**

| Method | Metric | 0000 | 0059 | 0106 | 0169 | 0181 | 0207 | Avg. |
|---|---|---|---|---|---|---|---|---|
| NICE-SLAM [34] | PSNR ↑ | 18.71 | 16.55 | 17.29 | 18.75 | 15.56 | 18.38 | 17.54 |
| | SSIM ↑ | 0.641 | 0.605 | 0.646 | 0.629 | 0.562 | 0.646 | 0.621 |
| | LPIPS ↓ | 0.561 | 0.534 | 0.510 | 0.534 | 0.602 | 0.552 | 0.548 |
| Vox-Fusion [30] | PSNR ↑ | 19.06 | 16.38 | 18.46 | 18.69 | 16.75 | 19.66 | 18.17 |
| | SSIM ↑ | 0.662 | 0.615 | 0.753 | 0.650 | 0.666 | 0.696 | 0.673 |
| | LPIPS ↓ | 0.515 | 0.528 | 0.439 | 0.513 | 0.532 | 0.500 | 0.504 |
| ESLAM [5] | PSNR ↑ | 15.70 | 14.48 | 15.44 | 14.56 | 14.22 | 17.32 | 15.29 |
| | SSIM ↑ | 0.687 | 0.632 | 0.628 | 0.656 | 0.696 | 0.653 | 0.658 |
| | LPIPS ↓ | 0.449 | 0.450 | 0.529 | 0.486 | 0.482 | 0.534 | 0.488 |
| Point-SLAM [19] | PSNR ↑ | 21.30 | 19.48 | 16.80 | 18.53 | 22.27 | 20.56 | 19.82 |
| | SSIM ↑ | 0.806 | 0.765 | 0.676 | 0.686 | 0.823 | 0.750 | 0.751 |
| | LPIPS ↓ | 0.485 | 0.499 | 0.544 | 0.542 | 0.471 | 0.544 | 0.514 |
| SplaTAM [6] | PSNR ↑ | 19.33 | 19.27 | 17.73 | 21.97 | 16.76 | 19.80 | 19.14 |
| | SSIM ↑ | 0.660 | 0.792 | 0.690 | 0.776 | 0.683 | 0.696 | 0.716 |
| | LPIPS ↓ | 0.438 | 0.289 | 0.376 | 0.281 | 0.420 | 0.341 | 0.358 |
| GS³LAM (Ours) | PSNR ↑ | 23.02 | 20.96 | 22.37 | 25.85 | 20.58 | 24.39 | 22.86 |
| | SSIM ↑ | 0.852 | 0.858 | 0.872 | 0.890 | 0.855 | 0.878 | 0.868 |
| | LPIPS ↓ | 0.277 | 0.213 | 0.205 | 0.189 | 0.252 | 0.195 | 0.222 |

**Table 2: Rendering performance on Replica [21].**

| Method | Metrics | R0 | R1 | R2 | O0 | O1 | O2 | O3 | O4 | Avg. |
|---|---|---|---|---|---|---|---|---|---|---|
| NICE-SLAM [34] | PSNR ↑ | 22.12 | 22.47 | 24.52 | 29.07 | 30.34 | 19.66 | 22.23 | 24.94 | 24.42 |
| | SSIM ↑ | 0.689 | 0.757 | 0.874 | 0.874 | 0.886 | 0.797 | 0.801 | 0.856 | 0.809 |
| | LPIPS ↓ | 0.330 | 0.271 | 0.208 | 0.229 | 0.181 | 0.235 | 0.209 | 0.198 | 0.233 |
| Vox-Fusion [30] | PSNR ↑ | 22.39 | 22.36 | 23.92 | 27.79 | 29.83 | 20.33 | 23.47 | 25.21 | 24.41 |
| | SSIM ↑ | 0.683 | 0.751 | 0.798 | 0.857 | 0.876 | 0.794 | 0.803 | 0.847 | 0.801 |
| | LPIPS ↓ | 0.303 | 0.269 | 0.234 | 0.241 | 0.184 | 0.243 | 0.213 | 0.199 | 0.236 |
| ESLAM [5] | PSNR ↑ | 25.32 | 27.77 | 29.08 | 33.71 | 30.20 | 28.09 | 28.77 | 29.71 | 29.08 |
| | SSIM ↑ | 0.875 | 0.902 | 0.932 | 0.960 | 0.923 | 0.943 | 0.948 | 0.945 | 0.929 |
| | LPIPS ↓ | 0.313 | 0.298 | 0.248 | 0.184 | 0.228 | 0.241 | 0.196 | 0.204 | 0.336 |
| Co-SLAM [25] | PSNR ↑ | 27.27 | 28.45 | 29.06 | 34.14 | 34.87 | 28.43 | 28.76 | 30.91 | 30.24 |
| | SSIM ↑ | 0.910 | 0.909 | 0.932 | 0.961 | 0.969 | 0.938 | 0.941 | 0.955 | 0.939 |
| | LPIPS ↓ | 0.324 | 0.294 | 0.266 | 0.209 | 0.196 | 0.258 | 0.229 | 0.236 | 0.252 |
| Point-SLAM [19] | PSNR ↑ | 32.40 | 34.08 | 35.50 | 38.26 | 39.16 | 33.99 | 33.48 | 33.49 | 35.17 |
| | SSIM ↑ | 0.974 | 0.977 | 0.982 | 0.983 | 0.986 | 0.960 | 0.960 | 0.979 | 0.975 |
| | LPIPS ↓ | 0.113 | 0.116 | 0.111 | 0.100 | 0.118 | 0.156 | 0.132 | 0.142 | 0.124 |
| GS-SLAM [29] | PSNR ↑ | 31.56 | 32.86 | 32.59 | 38.70 | 41.17 | 32.36 | 32.03 | 32.92 | 34.27 |
| | SSIM ↑ | 0.968 | 0.973 | 0.971 | 0.986 | 0.993 | 0.978 | 0.970 | 0.968 | 0.975 |
| | LPIPS ↓ | 0.094 | 0.075 | 0.093 | 0.050 | 0.033 | 0.094 | 0.110 | 0.112 | 0.082 |
| SplaTAM [6] | PSNR ↑ | 32.86 | 33.89 | 35.25 | 38.26 | 39.17 | 31.97 | 29.70 | 31.81 | 34.11 |
| | SSIM ↑ | 0.980 | 0.970 | 0.980 | 0.980 | 0.980 | 0.970 | 0.950 | 0.950 | 0.970 |
| | LPIPS ↓ | 0.070 | 0.100 | 0.080 | 0.090 | 0.090 | 0.100 | 0.120 | 0.150 | 0.100 |
| GS³LAM (Ours) | PSNR ↑ | 33.67 | 35.80 | 35.96 | 40.28 | 41.21 | 34.30 | 34.27 | 34.59 | 36.26 |
| | SSIM ↑ | 0.986 | 0.989 | 0.990 | 0.993 | 0.994 | 0.988 | 0.990 | 0.983 | 0.989 |
| | LPIPS ↓ | 0.051 | 0.039 | 0.046 | 0.040 | 0.030 | 0.065 | 0.061 | 0.081 | 0.052 |

**Table 3: Tracking performance on Replica [21] (ATE RMSE ↓ [cm]).**

| Method | R0 | R1 | R2 | O0 | O1 | O2 | O3 | O4 | Avg. |
|---|---|---|---|---|---|---|---|---|---|
| NICE-SLAM [34] | 0.97 | 1.31 | 1.07 | 0.88 | 1.00 | 1.06 | 1.10 | 1.13 | 1.07 |
| Vox-Fusion [30] | 1.37 | 4.70 | 1.47 | 8.48 | 2.04 | 2.58 | 1.11 | 2.94 | 3.09 |
| ESLAM [5] | 0.71 | 0.70 | 0.52 | 0.57 | 0.55 | 0.58 | 0.72 | 0.63 | 0.63 |
| Co-SLAM [25] | 0.65 | 1.13 | 1.43 | 0.55 | 0.50 | 0.46 | 1.40 | 0.77 | 0.86 |
| Point-SLAM [19] | 0.61 | 0.41 | 0.37 | 0.38 | 0.48 | 0.54 | 0.69 | 0.72 | 0.53 |
| GS-SLAM [29] | 0.48 | 0.53 | 0.33 | 0.52 | 0.41 | 0.59 | 0.46 | 0.70 | 0.50 |
| SplaTAM [6] | 0.31 | 0.40 | 0.29 | 0.47 | 0.27 | 0.29 | 0.32 | 0.55 | 0.36 |
| GS³LAM (Ours) | 0.27 | 0.25 | 0.28 | 0.67 | 0.21 | 0.33 | 0.30 | 0.65 | 0.37 |

### 4.3 Tracking Evaluation

Table 3 presents a comparison of the tracking performance between GS³LAM and state-of-the-art NeRF-based and 3DGS-based SLAM systems on the Replica [21] dataset. Since these methods employ a frame-to-model tracking strategy, SG-Field can more accurately represent the scene compared to NeRF-based methods, thus resulting in higher tracking precision. In contrast to SplaTAM [6], although semantic embedding allows GS³LAM to focus more on the edges and details of the field, tracking relies on prominent features rather than details, thereby leading to a slight decrease in accuracy.

### 4.4 Semantic Reconstruction Evaluation

Table 4 presents the quantitative comparison between GS³LAM and several contemporary neural semantic SLAM approaches. Following the protocol outlined in NIDS-SLAM [3], we report the mean Intersection over Union (mIoU) across four scenes from the Replica dataset [21]. From Table 4, it can be observed that leveraging SG-Field for semantic feature embedding within GS³LAM leads to noticeable enhancements (increased by 9.22%) when compared with competing NeRF-based semantic methods.

**Table 4: Semantic reconstruction accuracy on Replica [21] (mIoU ↑ [%]).**

| Method | Room 0 | Room 1 | Room 2 | Office 0 | Avg. |
|---|---|---|---|---|---|
| NIDS SLAM [3] | 82.45 | 84.08 | 76.99 | 85.94 | 82.37 |
| DNS SLAM [9] | 88.32 | 84.90 | 81.20 | 84.66 | 84.77 |
| SNI-SLAM [32] | 88.42 | 87.43 | 86.16 | 87.63 | 87.41 |
| GS³LAM (Ours) | **96.83** | **96.68** | **96.40** | **96.61** | **96.63** |

### 4.5 Ablation Study

***Cumulative Opacity Mask Ablation.*** As evidenced by the ablation experiments in Table 5, the utilization of cumulative opacity masks is pivotal within the GS³LAM framework. During the tracking stage, the observable region mask $M_{obs}$ serves as the basis for decoupling camera pose estimation and SG-field optimization. It prevents the influence of unoptimized SG-Field on the current frame's tracking, reducing tracking errors from 43.12cm to 0.21cm. In mapping, compared to randomly sampling pixels from the current frame for expansion, the unobserved region mask $M_{unobs}$ filters regions already optimized, thereby avoiding the addition of new Gaussians to regions already represented by Gaussians. This effectively controls the number of Gaussians in the field and improves PSNR by 2.22 dB and mIoU by 6.91%.

***DSR Ablation.*** As depicted in Fig. 6, the absence of DSR strategy results in the emergence of numerous Gaussians with large scales at scene edges or unobserved regions, leading to blurriness at object boundaries and spatial misalignment between geometry and semantics. Furthermore, as shown in Table 5, the clear geometric contours achieved by DSR can reduce tracking errors by 16%, increase PSNR by 1.17 dB, and enhance semantic reconstruction accuracy by 3.36%.

***RSKM Ablation.*** As illustrated in Fig. 7, our proposed RSKM achieves a 5.49 dB increase in PSNR while reducing the variance by 24.42 times, mitigating the optimization bias of the SG-Field and ensuring consistency in rendering across all perspectives. When

RSKM is not employed (using LCKM instead), in regions with a high number of co-observed frames, the SG-Field undergoes repetitive optimization across frames, making it challenging to converge. Conversely, in regions with fewer co-observed frames, under-optimization occurs due to insufficient sampling. Consequently, LCKM results in numerous holes and blurriness in the field. Furthermore, as indicated in Table 5, RSKM also reduces tracking errors and enhances semantic reconstruction accuracy, contributing tremendously to achieving a globally consistent map in terms of geometry, semantics, and appearance.

**Table 5: The ablation study on Replica "Office 1".**

| Method | Metrics | | | | | |
|---|---|---|---|---|---|---|
| | PSNR ↑ | SSIM ↑ | LPIPS ↓ | Depth [cm] ↓ | ATE [cm] ↓ | mIoU [%] ↑ |
| w/o $M_{obs}$ | 19.63 | 0.720 | 0.493 | 13.31 | 43.12 | 30.23 |
| w/o $M_{unobs}$ | 39.10 | 0.986 | 0.068 | 1.08 | 0.28 | 90.44 |
| w/o DSR | 40.04 | 0.990 | 0.059 | 0.85 | 0.25 | 93.96 |
| w/o RSKM | 37.48 | 0.983 | 0.081 | 1.23 | 0.29 | 89.13 |
| Ours | **41.21** | **0.993** | **0.046** | **0.41** | **0.21** | **97.35** |

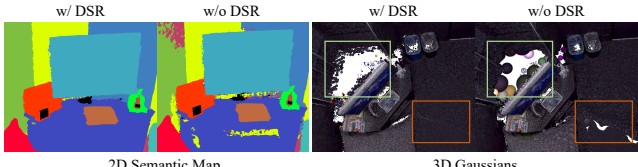

Figure 6: The ablation study of DSR on Replica "Office 1".

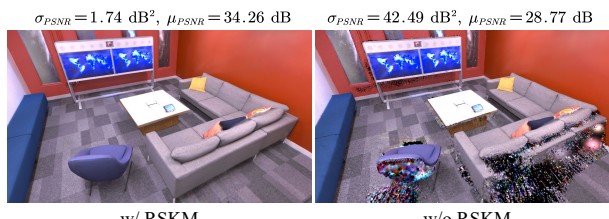

Figure 7: The ablation study of RSKM on Replica "Office 3".

## 5 CONCLUSION

We propose GS³LAM, a Gaussian Semantic Splatting SLAM system that utilizes 3D semantic Gaussians for dense map construction and tracking. Leveraging semantic Gaussian field scene representation, our approach better captures appearance, geometry, and semantics within the scene. Additionally, our proposed depth-adaptive scale regularization strategy adaptively adjusts the scales of Gaussians to characterize the scene, reducing uncertainties of 3D Gaussians at object surfaces and edges, thereby enhancing the accuracy of the 3D scene representation and achieving spatial alignment between geometry and semantics. Moreover, our proposed simple yet powerful random sampling-based keyframe mapping strategy effectively reduces optimization biases, mitigates the exacerbation of the forgetting phenomenon induced by semantic feature embedding, and enhances the global consistency of the semantic map. Thorough evaluations on benchmark datasets corroborate that GS³LAM outperforms its rivals noticeably in terms of tracking accuracy, rendering quality and speed, and semantic reconstruction.

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
