# OpenReview forum: "GS$^{3}$LAM: Gaussian Semantic Splatting SLAM"
_acmmm.org/ACMMM/2024/Conference — MM2024 Poster_

### Official Review · Reviewer_w9HA · 2024-05-19

**Rating:** 4
**Confidence:** 3

**Summary:**

Traditional semantic SLAM methods using explicit representations are limited in resolution and predictive capabilities. Inspired by the Gaussian Splatting, this paper introduced GS3LAM, a framework that inputs multimodal data to produce real-time, consistent, dense semantic maps. It features a Semantic Gaussian Field for scene modeling, Depth-adaptive Scale Regularization for alignment accuracy, and Random Sampling-based Keyframe Mapping for optimization. Extensive testing shows GS3LAM leads in tracking, rendering, and semantic reconstruction.

**Strengths:**

1. Method Innovation: This paper presents a pioneering approach to SLAM through the innovative use of a 3D semantic Gaussian representation. Furthermore, the paper introduces two key enhancements to the SLAM system: the "Depth-adaptive Scale Regularization (DSR) scheme" and the "Random Sampling based Keyframe Mapping (RSKM) strategy" to enhance the precision of the 3D scene representation.
2. Experiments Analysis: The paper has undergone a thorough evaluation process, which has conclusively demonstrated the superior performance of GS3LAM in comparison to other SOTA systems. The systematic ablation analysis further delves into the individual components of the SLAM framework, meticulously highlighting the contribution and effectiveness of each element.

**Limitations:**

1. Expanded Tracking Comparison: While the paper has established the method's superiority in rendering and tracking, a comparative analysis of tracking performance with other semantic NeRF/GS-based SLAM methods like NIDS SLAM, DNS SLAM, and SNI-SLAM would be beneficial. This would offer a more holistic view of the method's capabilities and standing within the field.
2. Localization Accuracy in Real-world Scenarios: The tracking results presented in Table 4 of the Supplementary Materials indicate a lower localization accuracy for the proposed method in real-world datasets like ScanNet. The authors should explain this performance to offer context and potential reasons for the observed results.
3. Memory Usage Comparison: Memory usage is a vital parameter, especially for systems that may be deployed on devices with limited resources. It is recommended that the authors include a comparison of memory usage to provide a complete picture of the system's resource requirements and efficiency.
4. Clarification on Mathematical Expressions: The notation "II" in Equation (12) appears to be non-standard and may be a typographical error. For precise understanding and application of the methodology, it would be essential for the authors to provide a clear definition or correct any typos to ensure the equations are interpreted correctly.

**Suitability:**

2

---

### Official Review · Reviewer_3Teb · 2024-05-19

**Rating:** 4
**Confidence:** 3

**Summary:**

In this paper, authors propose a Gaussian Splatting Semantic SLAM framework(GS3LAM) including two main modules. The first Depth-adaptive Scale Regularization (DSR) module constrains scales within a depth-dependent range to align Gaussians with geometric surfaces. The second Random Sampling-based Keyframe Mapping (RSKM) module is presented to address the forgetting phenomenon in GS3LAM. Extensive experiments conducted on Replica and ScanNet datasets demonstrate that our GS3LAM outperforms its counterparts in terms of tracking accuracy, rendering quality and speed, and semantic reconstruction.

**Strengths:**

Extend 3DGS as a semantic scene representation is interesting and urgent research topic.
Overall, the motivations for the two main innovations proposed by the authors are sufficient and reasonable.

**Limitations:**

The experimental section still needs improvement and refinement. 1、The author compares the open-source Gaussian schemes, but can add some works that have already been published in journals. 2、most important, In terms of algorithm superiority, the best and overall performance do not surpass the two Gaussian open-source works being compared. 3 The author claims that the Gaussian-based scheme is superior in time performance compared to the NeRF-based scheme, but this is not reflected in the experimental section.

**Suitability:**

3

---

### Official Review · Reviewer_9mrs · 2024-05-20

**Rating:** 4
**Confidence:** 4

**Summary:**

This paper introduces a novel semantic gaussian splatting SLAM system named GS3LAM. The system utilizes 3D semantic gaussian representation and a differentiable splatting rasterization pipeline to jointly optimize camera poses and fields for appearance, geometry, and semantics, achieving robust tracking, real-time high-quality rendering, and precise 3D semantic reconstruction.

**Strengths:**

- The paper introduces a new approach to SLAM by utilizing a 3D semantic Gaussian representation combined with a differentiable splatting rasterization pipeline. By integrating RGB, depth, and semantic information, this proposed GS3LAM can consistent and continuous dense semantic maps in real-time.
- The “Depth-adaptive Scale Regularization (DSR) scheme” and “Random Sampling based Keyframe Mapping (RSKM) strategy” help the SLAM system to better represent the scene and mitigate the forgetting phenomenon.
- Comprehensive evaluations have demonstrated GS3LAM's clear advantages over competing systems. The ablation analysis also illustrates the effectiveness of each component of the SLAM system.

**Limitations:**

Comments on the Methods:
- Some methodological formulas are not proposed by the author, such as Eq.19 and Eq.21. The relevant references should be added.
- The expressions in some Equations are irregular. What does the formulars “II” mean in Eq. (12)?
Comments on the Experiments:
- Lack of tracking comparison with semantic NeRF/GS-based SLAM methods such as NIDS SLAM, DNS SLAM and SNI-SLAM.
- In the comparison experiment of Rendering and Tracking, the superiority of this method is proved, but the comparison of Reconstruction results is missing.
- The Table4 in the Supplementary Materials shows the tracking results of the in the real-world dataset, ScanNet. Obviously, the localization accuracy of the proposed method in the real scene is lower than the others. The explanation should be provided.
- Runtime Analysis is a very important indicator in the SLAM system. According to the comparison results in Table1 of Supplementary Materials, GS3LAM has no advantage in the overall running efficiency, only the rendering speed is faster.
- It is recommended that the author give a comparison of Memory Usage parameters.
Other comments:
- Miswriting "known" as "konwn" in Abstract.
- PSNR in Figure1 should be the result of RGB rendering, but “Ours” is a semantic image.

**Suitability:**

2

---

### Official Review · Reviewer_qyZT · 2024-05-24

**Rating:** 4
**Confidence:** 2

**Summary:**

This paper introduces a new SLAM (Simultaneous Localization and Mapping) framework that combines semantic understanding with 3D Gaussian splatting for better tracking, high-quality rendering, and accurate semantic reconstruction. It uses Semantic Gaussian Fields (SG-Fields) and improves performance with depth-adaptive scale regularization (DSR) and random sampling-based keyframe mapping (RSKM).

**Strengths:**

The integration of semantic understanding into SLAM using 3D Gaussian splatting effectively captures appearance, geometry, and semantics.

The proposed DSR scheme improves alignment between geometry and semantics, reducing blurring and enhancing accuracy.

The proposed RSKM address the optimization bias and the forgetting phenomenon. Visual results show that RSKM enhances the consistency and accuracy of the semantic map, reducing artifacts and blurriness.

The extensive experiments conducted reveal that GS3LAM outperforms state-of-the-art methods in terms of tracking robustness, rendering quality, and semantic reconstruction precision.

**Limitations:**

The paper does not discuss other recent published works on Gaussian Splatting SLAM, such as [1], which may offer relevant insights and benchmarks.

The paper introduces the problem of forgetting early in the introduction but delays its definition and explanation until Method 3 and Figure 4. For better comprehension, the problem should be defined and explained earlier in the paper.

While the paper conducts extensive experiments on the ScanNet and Replica datasets, it lacks evaluations on additional datasets. The referenced works [1], [2], [3], [4], and [5] have included experiments on at least one or two other datasets, such as TUM-RGBD, in addition to ScanNet and Replica. This makes the experimental validation in this paper appear somewhat limited. To enhance the robustness and comprehensiveness of the evaluation, it is recommended to include results on the TUM-RGBD dataset.

Reference:

[1] Gaussian Splatting SLAM (CVPR 2024)

[2] SplaTAM: Splat, Track & Map 3D Gaussians for Dense RGB-D SLAM

[3] Point-SLAM: Dense Neural Point Cloud-based SLAM

[4] Vox-Fusion: Dense Tracking and Mapping with Voxel-based Neural Implicit Representation

[5] NICE-SLAM: Neural Implicit Scalable Encoding for SLAM

**Suitability:**

3

---

### Meta-Review · Area_Chair_iYha · 2024-07-02

**Recommendation:** Accept (Poster)
**Confidence:** 5

**Metareview:**

The paper receives mixed ratings from three reviewers. Two reviewers recommended positive recommendations. They acknowledged the technical novelty and the performance of the method. The second reviewer first gave a borderline accept and then changed the rating to borderline reject after the rebuttal. The major concern is that the rebuttal does not answer the question related to overall performance comparison to the competing methods, including both accuracy and efficiency. AC double-checked the experimental section and found that the comparison to the existing GS-based and NeRF-based methods shows clear improvements. Considering the positive majority of comments, AC finally recommends acceptance of this submission.